# STRATA: SIMPLE, GRADIENT-FREE ATTACKS FOR MODELS OF CODE

## ABSTRACT

Adversarial examples are imperceptible perturbations in the input to a neural model that result in misclassification. Generating adversarial examples for source code poses an additional challenge compared to the domains of images and natural language, because source code perturbations must adhere to strict semantic guidelines so the resulting programs retain the functional meaning of the code. We propose a simple and efficient gradient-free method for generating state-of-the-art adversarial examples on models of code that can be applied in a white-box or black-box setting. Our method generates untargeted and targeted attacks, and empirically outperforms competing gradient-based methods with less information and less computational effort.

## 1 INTRODUCTION

Although machine learning has been shown to be effective at a wide variety of tasks across computing, statistical models are susceptible to *adversarial examples*. Adversarial examples, first identified in the continuous domain by Szegedy et al. (2014), are imperceptible perturbations to input that result in misclassification. Researchers have developed effective techniques for adversarial example generation in the image domain (Goodfellow et al., 2015; Moosavi-Dezfooli et al., 2017; Papernot et al., 2016a) and in the natural language domain (Alzantot et al., 2018; Belinkov & Bisk, 2018; Cheng et al., 2020; Ebrahimi et al., 2018; Michel et al., 2019; Papernot et al., 2016b), although work in the source code domain is less extensive (see Related Work). The development of adversarial examples for deep learning models has progressed in tandem with the development of methods to make models which are robust to such adversarial attacks, though much is still being learned about model robustness (Goodfellow et al., 2015; Madry et al., 2018; Shafahi et al., 2019; Wong et al., 2019).

The threat of adversarial examples poses severe risks for ML-based malware defenses (Al-Dujaili et al., 2018; Grosse et al., 2016; Kaur & Kaur, 2015; Kolosnjaji et al., 2018; Kreuk et al., 2019; Suciu et al., 2019), and introduces the ability of malicious actors to trick ML-based code-suggestion tools to suggest bugs to an unknowing developer (Schuster et al., 2020). Thus, developing state-of-the-art attacks and constructing machine learning models that are robust to these attacks is important for computer security applications. Generating adversarial examples for models of code poses a challenge compared to the image and natural language domain, since the input data is discrete and textual and adversarial perturbations must abide by strict syntactical rules and semantic requirements. The CODE2SEQ model is a state-of-the-art model of code that has been used to explore adversarial example design and robustness methods on models of code (Rabin & Alipour, 2020; Ramakrishnan et al., 2020).

In this work, we propose the Simple TRAined Token Attack (STRATA), a novel and effective method for generating black-box and white-box adversarial attacks against CODE2SEQ. Our method replaces local variable names with high impact candidates that are identified by dataset statistics. It can also be used effectively for targeted attacks, where the perturbation targets a specific (altered) output classification. Further, we demonstrate that adversarial training, that is, injecting adversarial examples into CODE2SEQ's training set, improves the robustness of CODE2SEQ to adversarial attacks.

We evaluate STRATA on CODE2SEQ, though we hypothesize that the method can be applied to other models. The principles underlying STRATA apply not only to models of source code, but also to natural language models in contexts where the vocabulary is large and there is limited training data.

STRATA has a number of advantages compared to previously proposed adversarial attack strategies:

1. STRATA constructs state-of-the-art adversarial examples using a gradient-free approach that outperforms gradient-based methods;
2. STRATA generates white-box adversarial examples that are extremely effective; black-box attacks that use dictionaries created from unrelated code datasets perform similarly (Appendix C)
3. STRATA does not require the use of a GPU and can be executed more quickly than competing gradient-based attacks (Appendix D.1);
4. STRATA is the only available method (known to the authors at present) which performs targeted attacks on CODE2SEQ, which is the current state-of-the-art for models of code.

## 2 MOTIVATION

CODE2SEQ, developed by Alon et al. (2019a), is an encoder-decoder model inspired by SEQ2SEQ (Sutskever et al., 2014); it operates on code rather than natural language. CODE2SEQ is the state-of-the-art code model, and therefore it represents a good target for adversarial attacks and adversarial training. The model is tasked to predict method names from the source code body of a method. The model considers both the structure of an input program's Abstract Syntax Trees (ASTs) as well as the tokens corresponding to identifiers such as variable names, types, and invoked method names. To reduce the vocabulary size, identifier tokens are split into subtokens by commonly used delimiters such as camelCase and under_scores. In this example, subtokens would include "camel" and "case" and "under" and "scores". CODE2SEQ encodes subtokens into distributed embedding vectors. These subtoken embedding vectors are trained to capture semantic structure, so nearby embedding vectors should correspond to semantically similar subtokens (Bengio et al., 2003). In this paper, we distinguish between *subtoken* embedding vectors and *token* embedding vectors. *Subtoken* embedding vectors are trained model parameters. *Token* embedding vectors are computed as a sum of the embedding vectors of the constituent subtokens. If the token contains more than five subtokens, only the first five are summed, as per the CODE2SEQ architecture. The full description and architecture of the CODE2SEQ model is given in the original paper by Alon et al. (2019a).

The CODE2SEQ model only updates a subtoken embedding as frequently as that subtoken appears during training, which is proportional to its representation in the training dataset. However, the training datasets have very large vocabularies consisting not only of standard programming language keywords, but also a huge quantity of neologisms. The frequency at which subtokens appear in the CODE2SEQ java-large training set varies over many orders of magnitude, with the least common subtokens appearing fewer than 150 times, and the most common over $10^8$ times.

Thus, subtoken embedding vectors corresponding with infrequently-appearing subtokens will be modified by the training procedure much less often than common subtokens. Figure 1a demonstrates this phenomenon, showing a disparity between L2 norms of frequent and infrequently-appearing subtoken embedding vectors.

We confirm this empirically. When we initialized embedding vectors uniformly at random and then trained the model as normal, as per Alon et al. (2019a), we found that the vast majority of final, i.e., post-training, embedding vectors change very little from their initialization value. In fact, 90% of embedding tokens had an L2 distance of less than 0.05 between the initial vector and final, post-training vector when trained on a java dataset. About 10% of subtokens had a large L2 distance between the initial embedding and final embedding; these subtokens were more frequent in the training dataset and had embedding vectors with a notably larger final L2 magnitude (Figure 1).

The observation that high-L2-norm embedding vectors are associated with subtokens that appear sufficiently frequently in the dataset motivates the core intuitions of our attack[1]. We show in this paper that subtokens with high-L2-norm embedding vectors can be used for effective adversarial examples, which are constructed as follows:

---

[1]We note very-high-frequency subtokens have small L2 norms. Examples of these very-high-frequency subtokens include: `get`, `set`, `string`, and `void`, which appear so often as to not be useful for classification. Despite the fact that these subtokens are not good adversarial candidates for STRATA, there are so few of them that we expect them to have minimal influence on the effectiveness of our attack.

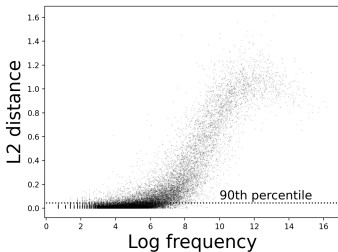
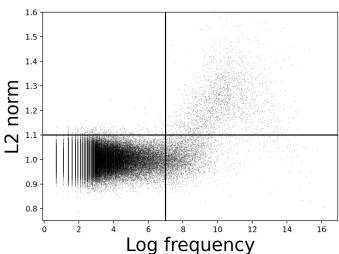

(a) The L2 distance between pre- and post-training embedding vectors for each subtoken, plotted against the log frequency of subtokens in the java-small training dataset.

(b) L2 norm of trained subtoken embedding vectors. Dashed and dotted lines chosen arbitrarily to illustrate tokens which have high L2 norm or frequency.

Figure 1: Visualization of embedding vectors for CODE2SEQ-SM.

1. To maximize adversarial effectiveness in a white-box setting, we should use tokens with high L2 norm embedding vectors as local variable name replacements. We confirm this empirically in the Experiments section.
2. In the absence of information about the L2 norms of embedding vectors, we can isolate high-L2-norm subtokens for local variable name replacement by selecting tokens which appear in the training dataset often enough to be well trained. This is empirically confirmed by the large intersection of high-L2-norm subtokens and subtokens with high frequency.

## 3 METHODS

### 3.1 DATASET DEFINITIONS AND CONSIDERATIONS

We evaluate our attack on four datasets that are used for training different CODE2SEQ models. There are three, non-overlapping Java datasets: java-small (700k examples), java-medium (4M examples), and java-large (16M examples) (Alon et al., 2019a), and one Python dataset, python150k (Raychev et al., 2016). We disambiguate the trained CODE2SEQ models for each datasets by denoting them CODE2SEQ-SM, CODE2SEQ-MD, CODE2SEQ-LG, and CODE2SEQ-PY for models trained on java-small, -medium, -large, and python150k respectively. Many of our experiments are evaluated on all four models; however, experiments that require adversarial training are only evaluated on CODE2SEQ-SM, for computational feasibility.

### 3.2 THE ATTACK

Traditional adversarial attacks on discrete spaces involve searching the discrete space for semantically similar perturbations that yield a misclassification. Searching the space of all possible valid discrete changes in source code is often intractable or even impossible (Rice, 1953). However, there are strategies to reduce the search space. For example, perturbations may be limited to a small number of predefined operations which are known to be semantically equivalent, such as inserting deadcode, replacing local variable names, or replacing expressions with known equivalent expressions. Gradient-based attacks on the embedding space may also be used in order to optimize the search of the space itself (Ramakrishnan et al., 2020; Yefet et al., 2020). However, gradient-based attacks are computationally expensive and rely heavily on knowledge of the exact parameters of the model.

We propose STRATA, which replaces local variable names with high-impact subtokens to generate adversarial examples. STRATA leverages the fact that only a relatively small number of subtoken embeddings are critical for the classification task performed by the CODE2SEQ model.

In Section 2, we presented two ways to identify high-impact subtokens. STRATA will use these to replace local variable names. Recall that the model composes these subtokens into tokens by summing the first five constituent subtoken embedding vectors. We wish to maximize the L2 norm of the resulting token, while minimizing the semantic change. We propose three strategies:

1. *single:* pick a single subtoken as the token;

2. *5-diff:* pick five different (not necessarily unique) subtokens and concatenate them, which will have a higher expected L2 norm than *single*;
3. *5-same:* pick a single subtoken, and repeat the subtoken five times to form a token, which will have the largest expected L2 norm, by the triangle inequality[2].

We subjectively propose that *single* is the smallest and most realistic semantic change, *5-same* is the largest change and the "best-case" for an adversarial example, and *5-diff* represents an intermediate attack strength.

For a given method, STRATA generates an untargeted perturbation as follows:

1. Select one random local variable $v$;
2. Choose an adversarial token $v^*$ appropriately, using the chosen concatenation strategy (*single*, *5-diff*, or *5-same*). For white-box attacks, choose each subtoken from a high-L2-norm vocabulary (top-$n$ by L2 norm). For black-box attacks, choose each subtoken with sufficiently high frequency (top-$n$ by frequency). We discuss the optimal cutoff values ($n$) for L2 and frequency in Section 3.4.
3. Replace $v$ with $v^*$.

For attacks on the Python dataset, since determining whether a variable is local or non-local is not always possible by looking at only the body of the method, we treat all variables as local.

To perform targeted attacks in which we want the output to include a particular subtoken $t$, we perform the same steps as the untargeted attack, and choose $v^*$ to be a *5-same* concatenation of $t$.

### 3.3 DATASET-AGNOSTIC BLACK-BOX ATTACKS

STRATA can generate effective adversarial attacks even without the training dataset. We can determine subtoken frequency statistics from a different (potentially non-overlapping) dataset. We empirically confirm that STRATA can use non-overlapping datasets (java-small, java-medium, and java-large) in this way to attack CODE2SEQ models that have been trained on a different Java dataset. We conclude that the subtoken distributions of non-overlapping Java datasets are sufficiently similar for STRATA to be effective (Appendix C). This confirms that STRATA can be applied in a completely black-box setting without knowledge of the model parameters or training dataset.

### 3.4 IMPLEMENTATION

We assayed the effectiveness of the method by applying a simple local variable token replacement attack. We select a single locally-declared variable at random and rename it with an adversarial token which does not conflict with another name. Because the initial variable is locally declared, we know that changing the name will have no effect elsewhere in the program, and will have no behavioral effect. The token replacement, however, can effectively attack CODE2SEQ; example shown in the Appendix.

The java-small test set consists of ~57,000 examples; we exclude from our testing dataset all methods that cannot be attacked by our method, i.e. all methods without local variables, leaving ~40,000 examples. The python150k dataset consists of 50,000 testing examples. We then create several vocabularies of subtokens from which to choose adversarial substitutions:

1. *All:* Contains all subtokens. Note that the number of subtokens varies by dataset (Table 1);
2. *Top $n$ by L2 norm* contains subtokens for which their L2 norm embedding vectors are the $n$ highest;
3. *Top $n$ by frequency* contains only the $n$ subtokens which occur in the training data with highest frequency.

To obtain optimal thresholds of $n$, we swept the range of possibilities to find the $n$ that minimizes F1 score, i.e., generates the best performing adversarial examples (Figure 2). We present the final values of $n$ in Table 1.

---

[2]The triangle inequality states that $\|\mathbf{x} + \mathbf{y}\| \leq \|\mathbf{x}\| + \|\mathbf{y}\|$, and is equal (and thus maximized) when $\mathbf{x}$ and $\mathbf{y}$ are colinear, which occurs when $\mathbf{x} = \mathbf{y}$. This is easily generalized to five vectors.

| Dataset | Model name | Optimal $n$ by frequency | Optimal $n$ by L2 | Total subtokens |
|---------|-----------|--------------------------|-------------------|-----------------|
| java-small | CODE2SEQ-SM | 1800 (2.51%) | 1000 (1.39%) | 71766 |
| java-med | CODE2SEQ-MD | 3000 (1.65%) | 3000 (1.65%) | 181125 |
| java-large | CODE2SEQ-LG | 10000 (5.54%) | 6000 (3.33%) | 180355 |
| python150k | CODE2SEQ-LG | 1600 (1.99%) | 1600 (1.99%) | 80336 |

Table 1: The name of the CODE2SEQ model trained on each dataset, along with optimal values of $n$ for the top-$n$ by frequency and L2 norm vocabularies, and the total number of subtokens for each dataset.

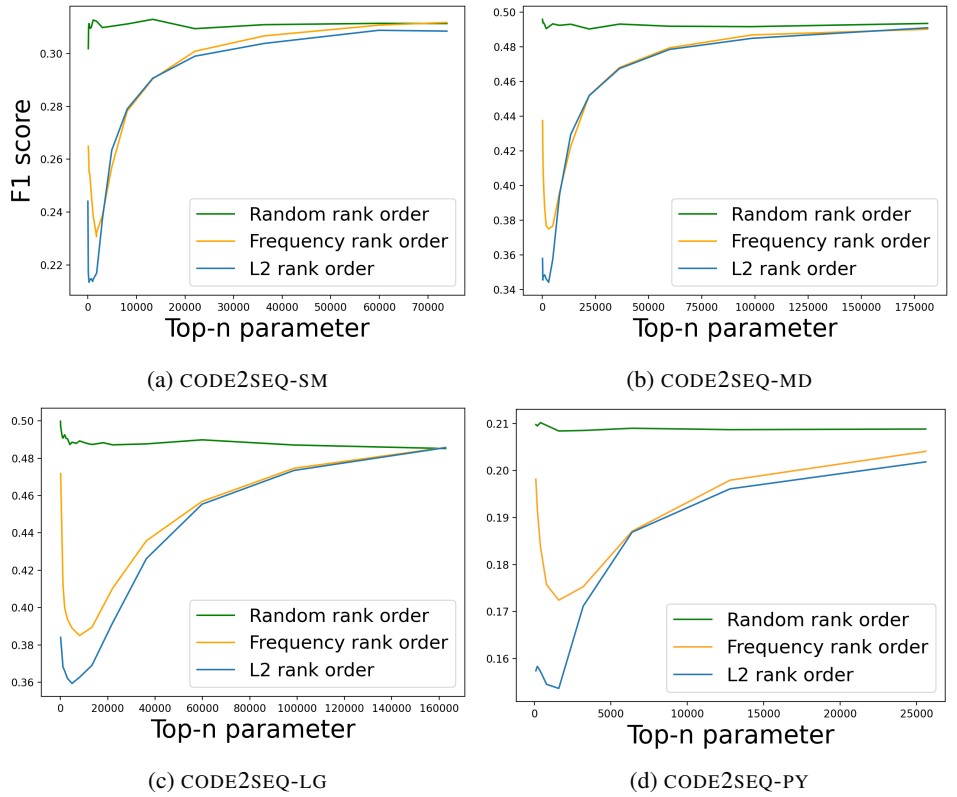

(a) CODE2SEQ-SM

(b) CODE2SEQ-MD

(c) CODE2SEQ-LG

(d) CODE2SEQ-PY

Figure 2: The F1 scores of the CODE2SEQ models, evaluated on the java-small or python testing dataset that has been adversarially perturbed with STRATA for given top-$n$ parameters, using *5-same* concatenation. A lower F1 score corresponds to a more effective attack.

## 4 EXPERIMENTS AND DISCUSSION

### 4.1 UNTARGETED ATTACKS

Table 2 compares the three proposed concatenation strategies on each of the four proposed vocabularies (three in Java, one in Python). To perform comparisons, we measure the F1 score of the model on the java-small testing dataset, with each adversarial perturbation. Lower F1 scores correspond with better attacks. We see that the performance of CODE2SEQ drops when a local variable is replaced with a token composed of a random selection from all subtokens, using the *5-diff* and *5-same* concatenation strategies. However, we observe a larger drop in F1 score when we make variable replacements from subtokens selected from the top-$n$ subtokens by L2 norm or by frequency, using values of $n$ specified in Table 1. When we replaced local variables with random tokens with high L2 norm, the F1 score dropped substantially, confirming our hypothesis that we can improve the effectiveness of adversarial examples by selecting replacement tokens such that their subtoken embedding vectors have high L2 norm. Similarly, adversarial examples that replace a local variable with subtokens of

high frequency in the training dataset are highly effective, suggesting that the black-box method of choosing adversarial subtokens based on frequency alone can approximate the white-box attack.

Surprisingly, the F1 score of CODE2SEQ-SM increased for random and frequency-based adversarial perturbations constructed with the *single* concatenation strategy, suggesting that CODE2SEQ-SM relies less on variable names for classification than CODE2SEQ-MD or CODE2SEQ-LG. The attacks on CODE2SEQ-PY were also incredibly effective, although the baseline accuracy for that model was already lower than the rest of the models.

| | Baseline | All | | | Top-$n$ by L2 norm | | | Top-$n$ by Frequency | | |
|---|---|---|---|---|---|---|---|---|---|---|
| | | *single* | *5-diff* | *5-same* | *single* | *5-diff* | *5-same* | *single* | *5-diff* | *5-same* |
| CODE2SEQ-SM | .369 | .381 | .350 | .310 | .362 | .263 | **.214** | .372 | .284 | **.231** |
| CODE2SEQ-MD | .564 | .548 | .531 | .492 | .513 | .416 | **.375** | .513 | .385 | **.345** |
| CODE2SEQ-LG | .608 | .536 | .547 | .488 | .542 | .396 | **.360** | .548 | .427 | **.388** |
| CODE2SEQ-PY | .313 | .536 | .547 | .488 | .249 | .198 | **.153** | .256 | .211 | **.172** |

Table 2: Evaluation of F1 scores of STRATA on our CODE2SEQ models, using vocabularies and replacement strategies as described in the Methods section. The baseline refers to the performance of the model on the original test dataset. For the Java models, samples were drawn from the java-small testing set such that each method included a local variable to perturb. All top-$n$ scores use optimal values of $n$ proposed in Table 1.

As proposed in Section 3.2, the effectiveness of the adversarial attack is optimized when we replace local variables with a token that is constructed with the *5-same* strategy. Strikingly, the black-box attack (top-$n$ by frequency) is nearly as effective as the white-box attack (top-$n$ by L2).

## 4.2 TARGETED ATTACKS

We perform targeted adversarial attacks on the Java datasets that aim to inject a chosen subtoken into the output by performing a STRATA attack using the targeted subtoken for replacement. See Appendix D for an example.

To assay the effectiveness of targeted attacks, we perform targeted attacks that target three different subtoken vocabularies: (1) all valid subtokens, (2) the optimized L2 vocabulary, and (3) the optimized frequency vocabulary, where vocabularies are optimized for CODE2SEQ-SM, CODE2SEQ-MD, or CODE2SEQ-LG appropriately. We determine that a particular attack is successful if the selected subtoken is included in the output. We measure the percent of successful attacks, thus computing an aggregate effectiveness of targeted attacks (Table 3). We omit targeted attacks on the Python dataset although we expect that a similar trend holds.

| Model | Perturbation | Strategy | % success |
|---|---|---|---|
| CODE2SEQ-LG | L2, top 6k | *5-same* | **37.1** |
| CODE2SEQ-LG | Frequency, top 10k | *5-same* | 35.6 |
| CODE2SEQ-LG | All | *5-same* | 3.9 |
| CODE2SEQ-MD | L2, top 3k | *5-same* | **43.8** |
| CODE2SEQ-MD | Frequency, top 3k | *5-same* | 39.1 |
| CODE2SEQ-MD | All | *5-same* | 1.4 |
| CODE2SEQ-SM | L2, top 1k | *5-same* | **58.7** |
| CODE2SEQ-SM | Frequency, top 1.8k | *5-same* | 52.8 |
| CODE2SEQ-SM | All | *5-same* | 2.1 |

Table 3: Effectiveness of targeted attacks on CODE2SEQ.

Table 3 reveals that CODE2SEQ is especially vulnerable to targeted attacks performed on high-impact subtokens. The black-box (frequency) attack performs similarly to the white-box (L2) attack.

## 4.3 TRANSFERABILITY

In this section we show that gradient-based adversarial training as proposed by Ramakrishnan et al. (2020) is not effective at improving robustness to STRATA attacks. We test a CODE2SEQ model that has been trained to be robust to the gradient-based adversarial examples proposed by Ramakrishnan et al. (2020), and find that the model is indeed robust to the gradient-based token-replacement perturbations. However, neither the original nor the robust model are impervious to perturbations produced by STRATA (Table 4). This result confirms that STRATA can effectively target models that are robust to some gradient-based perturbations; therefore it is a useful tool when hardening models of code, even when gradient-based perturbations are also being used.

| Model | No perturbations (F1) | Gradient perturbations (F1) | STRATA perturbations (F1) |
|---|---|---|---|
| Original | .363 | .243 | **.212** |
| Robust | .367 | .342 | **.240** |

Table 4: F1 scores of a non-robust and robust CODE2SEQ-SM model on gradient-based adversarial perturbations and STRATA perturbations.

## 5 STRATA OUTPERFORMS SIMILAR ATTACKS

At the time of writing there are two other works that address adversarial attacks targeting CODE2SEQ: a class of source code transformations by Rabin & Alipour (2020) and gradient-based attacks by Ramakrishnan et al. (2020). We greatly outperform the transformations considered in Rabin & Alipour (2020) (Appendix B).

We compare our work to gradient-based adversarial perturbations proposed by Ramakrishnan et al. (2020), in which they attack a CODE2SEQ-SM model. We consider the variable replacement, print statement insertion, try-catch statement insertion, and worst-case single transformation attacks for our comparison. Note: to establish a fair comparison, we include the 17,000 examples in the testing set that do not include a local variable, and thus we do not even attempt to perturb them, hence why the F1 scores of the adversarial examples are larger than as reported in Table 2. We find that STRATA outperforms all attacks performed by Ramakrishnan et al. (2020) except for the worst-case transformation, which is inherently a larger transformation than our varible-replacement attack. This includes greatly outperforming the gradient-based local variable replacement attack and performing similarly to the worst-case transformation, despite the fact that STRATA generates smaller perturbations with less computational effort, and with less information about the model (Table 5). These results indicate that STRATA attacks are state-of-the-art on CODE2SEQ.

| Adversarial attack | F1 | % of baseline score |
|---|---|---|
| Baseline (Ramakrishnan et al.) | .414 | 100 |
| Variable Replacement (Ramakrishnan et al.) | .389 | 93.9 |
| Try-catch Statement Insertion (Ramakrishnan et al.) | .336 | 81.2 |
| Print Statement Insertion (Ramakrishnan et al.) | .312 | 75.3 |
| Worst-case Transformation (Ramakrishnan et al.) | .246 | 59.4 |
| Baseline (STRATA) | .425 | 100 |
| Top 1k by L2 (STRATA) | **.316** | **74.4** |
| Top 1.8k by frequency (STRATA) | .328 | 77.2 |

Table 5: Comparison of F1 scores our method to gradient-based attacks described by Ramakrishnan et al. (2020). Both our model and the Ramakrishnan et al. (2020) model are trained on java-small. We perform *5-same* attacks.

## 6 RELATED WORK

**Adversarial examples for models of code** Allamanis et al. (2018) provide a comprehensive survey of prior research on models of code. Several papers develop techniques for generating adversarial examples on models of source code: Quiring et al. (2019) perform adversarial attacks on source code by applying a Monte-Carlo tree search over possible transformations, and Zhang et al. (2020) apply Metropolis-Hastings sampling to perform identifier replacement to create adversarial examples. Bielik & Vechev (2020) improve on the adversarial robustness of models of code by developing a model that can learn to abstain if uncertain.

**Adversarial examples for CODE2VEC and CODE2SEQ** Yefet et al. (2020) generate both gradient-based targeted and untargeted adversarial examples for CODE2VEC Alon et al. (2019b). Most directly related to our paper, Ramakrishnan et al. (2020) perform gradient-based adversarial attacks and adversarial training on CODE2SEQ. Rabin & Alipour (2020) evaluate the robustness of CODE2SEQ to semantic-preserving transformations.

**Targeted attacks on models of code** As previously noted, Yefet et al. (2020) propose a method for targeted adversarial examples for CODE2VEC (Alon et al., 2019b). To our knowledge, at the time of writing, no other paper performs targeted attacks on CODE2SEQ.

**Adversarial training for robustness** Many recent papers have examined the robustness of neural network models to adversarial perturbations. Szegedy et al. (2014) demonstrates that neural networks are vulnerable to adversarial perturbations, but that training on these perturbations, i.e., adversarial training, can increase robustness. Further papers explore faster and more effective methods of adversarial training to improve robustness, though mostly in the continuous domain (Madry et al., 2018; Shafahi et al., 2019; Wong et al., 2019). Ramakrishnan et al. (2020) perform adversarial training on CODE2SEQ to improve robustness. Yefet et al. (2020) propose multiple methods to improve robustness of models of source code, including adversarial training, outlier detection, and excluding variable names from model input.

## 7 CONCLUSION

In this work, we presented STRATA, a simple, gradient-free method for generating adversarial examples in discrete space that can be used to help build robustness in models of code. Because the L2 norm of an embedding vector can be approximated by the frequency in the training data, STRATA can be used to generate gradient-free white- and black-box attacks. We presented effective attacks using this method, including targeted attacks, and also showed that adversarial fine-tuning using STRATA examples can lead to increased robustness, even when the fine-tuned model is only being tested on clean data.

Our work does have some limitations. In the continuous domain, many adversarial attacks may not even be detected by humans; in the discrete domain, this is not possible, but some attacks in the discrete space are more realistic (harder to spot) than others. The most powerful attack we propose, *5-same*, is also the least realistic; *single* is a smaller perturbation but it has less effective attacks. We expect that the attack could be expanded by adding dummy local variables to methods that do not initially contain a local variable, mitigating the current inability to attack methods without local variables.

STRATA does not just have application as an attack, but also as a defense. We present preliminary adversarial training results in the appendix which suggest that using adversarial examples generated with STRATA for adversarial training can improve robustness to adversarial attacks. Since we show that standard gradient-based adversarial training is ineffective at defending against STRATA, it is important for robust models to be adversarial training with STRATA adversarial examples to ensure a comprehensive robustness. A complete study and evaluation of the effectiveness of adversarial training using STRATA is an important future direction.

Another exciting area of future inquiry would be the application of this attack to other models, including natural language models. The magnitude differences in subtoken frequency in code is similar to the magnitude differences in word frequency in natural language (Piantadosi, 2014). In fact,

tokens in software follow the same distribution as words in natural language (Zhang, 2008). Thus we believe that our technique should be applicable in the NLP setting (with modification; for example, replacement dictionaries might only be filled with high-impact synonyms). We theorize that STRATA could be used to good effect in models of natural language as well as models of code, and that the basic intuitions underlying the attack can extend even to domains where the actual training data is not available, as long as a suitably similar dataset is available. Our results stem from fundamental features of dataset distribution and training process, and so they should generalize to other applications.

STRATA has a broad application to the field of machine learning in discrete spaces with large vocabularies. We present the insight that subtoken frequency correlates with L2 norm and thus can be used to determine high impact subtokens; this technique is not *a priori* exclusive to code2seq or even models of code in general since it relies on properties of the dataset distribution and training technique and not the model architecture. STRATA is a high-impact, low-cost attack strategy that can be used to bolster robustness; we have made its code open-source so that those who wish to use it may do so.

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

## A    ADVERSARIAL TRAINING

We perform adversarial training in order to make CODE2SEQ most robust to adversarial attacks. To test the robustness of an adversarially-trained CODE2SEQ model, we perform the following experiment:

1. We train a CODE2SEQ-SM model.
2. We generate STRATA perturbations of the java-small training dataset using a vocabulary of the top-1000 subtokens by L2 norm of the embedding. For each training example, we generate five adversarial perturbations for each concatenation strategy, *single*, *5-diff*, and *5-same*.
3. We fine-tune the CODE2SEQ-SM model with the adversarial examples for a final epoch.

|          | Baseline | *single* | *5-diff* | *5-same* |
|----------|----------|----------|----------|----------|
| Clean-Tr | .369     | .362     | .263     | .214     |
| Adv-Tr   | **.371** | **.382** | **.370** | **.366** |

Table 6: Results of adversarial training of CODE2SEQ-SM on L2-based adversarial examples.

Our adversarial training results are preliminary. Further work is needed to evaluate the robustness of the model after adversarial training. Nonetheless, our results strongly suggest that adversarial training with STRATA adversarial examples is highly effective at defending against STRATA attacks (Table 6). Unexpectedly, training with adversarial examples slightly improves the ability of the model to classify clean data (improvement from F1 of 0.369 to 0.371), suggesting that the added robustness forces the model to learn a better representation of the source code. This adversarial fine-tuning was not more computationally intensive than standard training and represents a simple way to make this model of code more robust.

Our method differs from standard adversarial training method proposed in Madry et al. (2018) in which adversarial examples are regenerated at every epoch. Since our adversarial examples only rely

on knowing the top-$n$ subtokens by embedding L2 norm which does not change drastically over time, we do not have to recompute new adversarial examples after each epoch and instead can compute large batches of adversarial examples prior to training. Since STRATA is gradient-free and does not require re-computation every step in the adversarial training process, our method is extremely inexpensive, whereas traditional gradient-based methods for developing adversarial examples are highly expensive, and are less effective (Table 5). Therefore, our method can represent an extremely easy hardening technique for all types of models operating in the discrete domain with a large vocabulary space. Further, we have shown that traditional gradient-based adversarial training is largely ineffective at defending against our attack (Table 4), thus our method is vital for a comprehensive defense against adversarial attacks.

## B    FULL COMPARISON WITH OTHER CODE2SEQ ATTACKS

We have shown that STRATA works well to attack the CODE2SEQ model and can outperform the attacks by Ramakrishnan et al. (2020). Here, we present a comparison by the transformations proposed by Rabin & Alipour (2020). In order to be able to compare the same metrics, we calculate the percent prediction change, which is the percent of the time that an adversarial change resulted in a change in prediction. A higher percent prediction change indicates a better attack.

In Table 7, we compare the performance of our attack to the performance of the transformations generated by Rabin & Alipour (2020) We find that the transformations performed by Rabin & Alipour (2020) result in fewer prediction changes than STRATA. As above, the most effective strategy is our *5-same* attack.

| Adversarial attack | % prediction change |
|---|---|
| Variable renaming (Rabin et al.) | 47.04 |
| Boolean Exchange (Rabin et al.) | 51.43 |
| Loop Exchange (Rabin et al.) | 42.51 |
| Permute Statement (Rabin et al.) | 43.53 |
| All (STRATA, *5-same*) | 77.3 |
| **Top 6k by L2 (STRATA, *5-same*)** | **86.3** |
| Top 10k by Frequency (STRATA, *5-same*) | 84.5 |

Table 7: Comparison of the percent of the time the prediction is changed by STRATA and by an attack from *Rabin et al.* Both our attack and the attack from *Rabin et al.* target a CODE2SEQ-LG. Note that our model is evaluated on the java-small testing dataset.

## C    CROSS-DATASET ATTACKS

We present two fully black-box attacks that do not require any information about the targeted CODE2SEQ model or dataset:

As a surrogate model, we train a CODE2SEQ model on any available dataset for the targeted programming language. To obtain adversarial examples from the surrogate, we identify optimal L2 and frequency cutoffs for this model. Using these cutoffs, we construct a vocabulary of the optimal top-$n$ by frequency or by L2 norm. We show that these adversarial examples can be transferred to other models.

We present the results of the cross-dataset transfer attack proposed in Section 3.3. In particular, we generate both frequency and L2 STRATA adversarial examples. We use the L2 norm of the embeddings of CODE2SEQ-SM, CODE2SEQ-MD, and CODE2SEQ-LG, and the subtoken frequencies of java-small, java-medium, and java-large to construct six different collections of adversarial examples, of which each collection is a perturbation of the java-small test set. We test each dataset on each model. Tables 8 and 9 show the results of the experiments, revealing that while the white-box and known-dataset attacks (the diagonals of the tables) outperform the cross-dataset attacks, the cross-dataset attacks are nonetheless effective. Furthermore, we note that L2-based cross-dataset attacks are more effective than frequency-based cross-dataset attacks, confirming that L2 norms can

effectively identify subtokens that are high impact in other models. We conclude that STRATA can be performed in a true black-box setting with no information about the model parameters nor the training dataset. The cross-dataset attack is likely effective due to similar distributions of the Java datasets. Similar to word frequencies in natural language corpora, we expect that most Java datasets should have a similar subtoken distributions, and thus STRATA should transfer across models trained on different datasets.

|                | java-small (1k, L2) | java-medium (3k, L2) | java-large (6k, L2) |
|----------------|---------------------|----------------------|---------------------|
| CODE2SEQ-SM    | .214                | .253                 | .290                |
| CODE2SEQ-MD    | .359                | .349                 | .433                |
| CODE2SEQ-LG    | .373                | .361                 | .358                |

Table 8: F1 scores on adversarial data generated by a cross-dataset attack where vocabularies are constructed by using the L2 norm of the embeddings of the CODE2SEQ model trained on the particular dataset. The first column corresponds with the model that is being attacked, and the other columns correspond with the dataset from which the attack is constructed. Lower score means a better attack. All adversarial subtokens are concatenated with *5-same*. The boxed scores correspond with the baseline same-dataset attacks.

|                | java-small (1.8k, freq) | java-medium (3k, freq) | java-large (10k, freq) |
|----------------|-------------------------|------------------------|------------------------|
| CODE2SEQ-SM    | .235                    | .248                   | .279                   |
| CODE2SEQ-MD    | .381                    | .377                   | .417                   |
| CODE2SEQ-LG    | .397                    | .392                   | .387                   |

Table 9: F1 scores on adversarial data generated by a cross-dataset attack where vocabularies are constructed by using the frequency of subtokens in the associated training dataset. The first column corresponds with the model that is being attacked, and the other columns correspond with the dataset from which the attack is constructed. Lower score means a better attack. All adversarial subtokens are concatenated with *5-same*. The boxed scores correspond with the baseline same-dataset attacks.

## D  EXAMPLES OF TARGETED ATTACKS

```
public static int f(List<Integer> input) {
    int total = 0;
    for (int i=0; i<input.length; ++i) {
        total += input[i];
    }
    return total;
}
```
(a) Predicted as `sum`

```
public static int f(List<Integer> input) {
    int answer = 0;
    for (int i=0; i<input.length; ++i) {
        answer += input[i];
    }
    return answer;
}
```
(b) Predicted as `getInt`

Figure 3: Changing local variable identifiers affects prediction. *Left*: Original Java method `sum` computes the sum of a list of integers. CODE2SEQ, trained on java-large, predicts that the method name is `sum`. *Right:* Perturbed Java method with the same behavior, but the local variable `total` has been replaced with `answer`, causing CODE2SEQ to misclassify this method as `getInt`.

To illustrate the effectiveness of STRATA targeted attacks more concretely, we target particular arbitrarily-picked subtokens and measure the success rate over the entire testing set (Table 10) and find that though effectiveness can vary across different targets, the average effectiveness is quite high.

### D.1  COMPUTATIONALLY INEXPENSIVE

Current alternative methods for attacking models of source code with comparable results involve either an extensive search for optimal transformations or gradient-based optimization for token

| Target | % | Target | % | Target | % | Target | % |
|--------|------|--------|------|---------|------|----------|------|
| tournament | 86.3 | ftp | 53.4 | concat | 41.5 | combat | 26.6 |
| redaction | 70.5 | podam | 45.3 | eternal | 37.3 | thursday | 23.9 |
| outliers | 62.4 | wind | 44.7 | orderby | 32.8 | girth | 21.5 |
| mission | 56.8 | weld | 42.4 | reentry | 29.8 | ixor | 16.4 |

Table 10: Effectiveness of targeted attacks on CODE2SEQ-LG using uncurated and arbitrarily picked subtokens of high embedding vector, sorted by % success.

replacement, or a combination of the two (Rabin & Alipour, 2020; Ramakrishnan et al., 2020; Yefet et al., 2020). Extensive searches are inherently computationally expensive and, in the case of gradient-based optimization, oftentimes require a GPU for efficient implementation. STRATA, however, can be implemented to run quickly on even CPU-only machines. After an initial pre-processing step to mark local variable names for easy replacement which took less than five minutes on our 24-core CPU-only machine, we were able to construct adversarial examples using STRATA on a dataset of 20,000 within seconds. The analogous gradient-based method proposed by Ramakrishnan et al. (2020) took multiple hours on the same machine. The combined speed and effectiveness of STRATA will allow researchers to quickly harden their models against adversarial attacks with efficient large-scale adversarial training.

