# OpenReview forum: "STRATA: Simple, Gradient-free Attacks for Models of Code"
_ICLR.cc/2021/Conference — Reject_

### Official Review · AnonReviewer1 · 2020-10-25
**A simple adversarial attack against code2seq**

**Rating:** 4
**Confidence:** 4

**Review:**

This paper proposes STRATA, a simple adversarial attack against the code2seq model. The key idea is to replace local variable names in the input code with other randomly chosen sub-tokens with embedding vectors of relatively high L2 norms. Meanwhile, they observe that such tokens often appear frequently in the training set, thus alternatively they can simply use frequently appeared tokens as the target to perform the attacks. In this way, they can attack the model in the black-box scenario, without the knowledge of the model parameters and the training data, as long as they can roughly approximate the frequency distribution of different code sub-tokens in the training set. They evaluate their approach on code2seq models trained for the Java code, and compare with existing attacks against code2seq models. They first show that the 5-same attack, i.e., repeating a sub-token 5 times and concatenating them as the new local variable name, is the most effective attack. This attack decreases the F1 scores more compared to the baseline attack from prior work. In addition, they show that by adding STRATA adversarial examples for adversarial training, the new model becomes more robust to their proposed attacks.

Robustness of models of code is an interesting topic, and the authors show that a simple and computationally inexpensive attack could degrade the performance of the code2seq model. However, this paper suffers from both the approach design and the evaluation, and I discuss the details below.

1. A very straightforward defense is to anonymize the variable names. In particular, for 5-same attacks, although each sub-token itself is common, the concatenation of them doesn't seem natural, and thus a simple input preprocessing could already make the attack ineffective.

2. The authors mention that they select ~40,000 test samples from the original test set to perform the attacks. In this case, when computing the F1 score, is it computed among the 40,000 test samples, or on the entire test set? For results of attacks proposed in prior work, have you evaluated them yourself, or did you quote the numbers from previous papers? I see that the baseline results are the same as shown in Ramakrishnan et al., but I don't think they limit their attacks to code with local variables, thus I would like to double-check.

3. When evaluating the adversarial training and the transferability of the previous defense, they show that the previous defense does not show better robustness against STRATA. On the other hand, training with STRATA may also not improve the robustness against previous attacks. Have the authors evaluated the transferability of the model trained with STRATA adversarial examples? Meanwhile, did you train the same model to conduct attacks using STRATA and the baseline? I noticed that the F1 scores of models without perturbations are different for STRATA and the baseline attack.

4. The scope of experiments is pretty limited. The authors only focus on a single model (code2seq) for one task (Java). The authors should extend the experiments to more models and/or more code-related tasks, or at least different programming languages, such as those evaluated in the code2seq paper or previous papers on attacks.

---

> ### Author Response · Authors · 2020-11-14
> **Thank you for your feedback**
>
> My co-authors and I would like to thank you for compiling this thoughtful review of our work, STRATA. We would like to share the following response to your comments:
>
> * While anonymizing the variable names would potentially defend against these attacks, it would also make the baseline performance of the model suffer greatly because of its high dependence on token names (see Table 3 in the original code2seq paper by Alon et al.) Your point about the unrealistic nature of the 5-same attack is well-taken, however, and thus we included information about two more realistic attacks, 1-same and 5-diff. The 5-same attack represents a "best-case" for the attacker, since at most 5 subtokens are allowed in the input architecture for code2seq.
> * We did quote the metrics for comparison from the Ramakrishnan et al. and Rabin et al. papers. Your point that we only use testing examples with local variable names is a good one, and we will update the paper to include the F1 scores from the entire testing set in the next revision. Nonetheless, our attack still significantly outperforms other attacks, especially previous local variable replacement attacks.
> * We agree that testing whether or not adversarial training with STRATA adversarial examples can defend against previously proposed attacks is a good idea, thank you for this suggestion.
> * You are right that the scope of our experiments is, at present, limited. We are working on a revised version of the manuscript that includes several more tasks (languages) and potentially another model as well. Please refer to the meta-review where we discuss why we think STRATA represents a novel, fundamental strategy that can be useful cross-domains even on its current experimental grounding.
>
> Thanks again for your time reading the paper and preparing these comments. We appreciate the feedback greatly and look forward to presenting a revised edition in the coming weeks.

---

### Official Review · AnonReviewer3 · 2020-10-28
**Interesting idea, limited results**

**Rating:** 4
**Confidence:** 4

**Review:**

Summary:

This paper proposes STRATA, a novel adversarial attack against source code models, more precisely against code2seq. The attack strategy can be applied under black- or white-box threat models, targeted or untargeted. Adversarial training is based on STRATA adversarial examples is proposed to render the models robust. Experiments are performed on Java code datasets of variable sizes.

Strong points:
- STRATA is based on the interesting observation that token frequencies and their $L_2$ norms are strongly correlated. This information provides an elegant and effective strategy for choosing attack tokens.
- The proposed attack seems computationally inexpensive.
- The paper covers what one would expect when proposing a new attack, including performance evaluation, study of transferability and adversarial training results.
- The paper is clear and well-written.

Concerns:
- STRATA seems limited in some respects: it requires access to a similar code base, can only be applied for functions with local variables, only replaces variable names.
- The proposed strategies for generating replacement tokens are trivial and easily detectable (e.g., concatenating the same token five times). The more subtle strategies are significantly less effective.
- The proposed adversarial training strategy does not seem enough for producing a robust model: only one epoch of fine-tuning on adversarial samples is performed at the end of natural training. This is coupled with unconvincing experimental evaluations of adversarial training. As such, the paper does not seem to prove that a robust model can or has been trained using STRATA samples.
- The notion of adversarial examples against code in the sense used in the paper seems vague. Appendix Fig. 3 shows a qualitative example of attack, where "product" is replaced with "identity" in all variable names of a function. The model (incorrectly) predicts the name of the function being related to "identity", not "product". However, that prediction seems reasonable; a human would most likely predict the same, considering that nothing in the code of the function is related to "product". Can this indeed be considered an adversarial example?
- It is unclear why the comparison to previous attacks is done under different metrics depending on the attack. Moreover, why is STRATA trained on the large Java dataset, but evaluated on the small one (Appendix, Tab. 7)?

Reasons for score:

Overall, I lean towards rejection. I like the idea behind choosing attack tokens, but the experimental results and derivation of the adversarial training strategy do not seem thorough enough. Moreover, an attack designed specifically for models against code, tested on one architecture and one programming language only seems relevant to a small part of the community.

---

> ### Author Response · Authors · 2020-11-14
> **Thank you for your feedback**
>
> My co-authors and I would like to thank you for compiling this thoughtful review of our work, STRATA. We would like to share the following response to your comments:
>
> * We were surprised to hear that you thought the approach was limited; STRATA’s technique of utilizing token distributions in the training set is both novel and extremely broad. We want to be clear that we agree there are limitations in the current implementation, but while it is true that at present STRATA only replaces local variable names, we expect that the concept of selecting likely candidates for high impact replacements by looking at the L2 norm can be applied in a much more broad setting, including models of natural language (first selecting out for synonyms) as well as other token-replacement attacks for models of code. For example, we expect that the attack could be expanded by adding dummy local variables to methods that do not initially contain a local variable, mitigating the current inability to attack methods without local variables.
> * To address your concern that our method may rely on having a similar codebase, we expect that any codebase that shares the same programming language should suffice, since we expect that the frequency of subtokens follow at least a loosely similar distribution in most code. We expect that using free resources such as GitHub to get training data for the attack should be sufficient. We show that this is the case for the java-small, -medium, and -large datasets in the appendix, despite the fact that they are non-overlapping.
> * While the more subtle strategies (i.e., single-subtoken replacement as opposed to 5-same) are less effective, they are still significantly more effective than current state-of-the-art variable replacement techniques, as shown by our comparison with the work by Ramakrishnan et al. and Rabin et al.
> * You raise a good point about the validity of that example snippet. However, many of the code snippets are less trivial than that one; we will replace the example.
> * Thank you for raising your concern about the comparison to previous attacks. We chose to compare our method to the work by Ramakrishnan et al. and Rabin et al. using metrics that match the metrics used in their original papers so that the comparison is on even footing.
> * We initially chose to evaluate on the java-small testing dataset wherever possible, (even for the model trained on java-large) for consistency and comparison purposes, however, in the next revision we will include the results using the java-large testing dataset where applicable.
>
> Thank you again for your helpful commentary and critiques; we are currently in the process of revising to address these concerns.

---

> > ### Comment · AnonReviewer3 · 2020-11-24
> > **Thank you for your response**
> >
> > Thank you for taking the reviewers' comments on board and for amending the paper. I also appreciate your detailed response to all the points that were raised. I believe the edits have brought additional value to the paper.

---

### Official Review · AnonReviewer4 · 2020-10-29
**Review for STRATA: Building Robustness with a Simple Method for Generating Black-box Adversarial Attacks for Models of Code**

**Rating:** 5
**Confidence:** 3

**Review:**

The paper proposes a gradient-free method to craft adversarial examples for Code2Seq model. Code2Seq model generates textual summary of code snippets.

Overall I think authors proposed an interesting idea of an attack, however evaluation should be improved. Thus I recommend to reject paper at this point, but encourage authors to improve the paper and resubmit.

Strong points:
* Proposed novel interesting idea of gradient-free attack on source code
* Paper is well written and easy to understand

Weak points:
* My main concern is evaluation. While authors claim that they compared to Ramakrishnan et al, in practice it looks like this comparison was performed using different baselines. Which makes results questionable.
* The whole paper is about attacking one very specific model (Code2Seq) and it’s not clear what is the motivation of attacking this specific model (instead of some other). It’s also not clear how easy proposed method could be adapted to other tasks (which use code as an input).
* Adversarial training is done by generating adversarial examples only one time (before the last epoch). This may lead for the model to overfit to specific adversarial examples.
* No comparison of adversarial training with any cheaper mechanism. For example, adding random noise to embeddings during training.

Recommendations on how to improve the paper:
* Improve evaluation, add fair comparison (i.e. on the same baseline model) with other methods.
* Show how this method could be adapted to different tasks which use code as an input and add evaluation on these tasks.
* Update adversarial training to be consistent with what is typically done (i.e. adversarial examples generated on each training step for current state of the model) or provide motivation for the current way adversarial training is implemented.

---

> ### Author Response · Authors · 2020-11-14
> **Thank you for your feedback**
>
> My co-authors and I would like to thank you for compiling this thoughtful review of our work, STRATA. We would like to share the following response to your comments:
> * Your concern about the validity of our comparison with the Ramakrishnan et al. paper is a great point and well-taken. The difference in baseline model performance is due to us evaluating exclusively on methods that contain local variables, whereas Ramakrishnan et al. evaluate on all methods in the test set. In our upcoming revision, we will add the results of the STRATA attack on the entire testing dataset to compare with the same metric as Ramakrishnan et al.
> * We used code2seq due to the fact that it was state-of-the-art. You raise a good point that our motivation for choosing code2seq was not as clear as it could have been. We are revising that section to include more information about our choice, as well as information about the ease of adaptability to new contexts and models (which we expect to be very simple).
> * You make two very good points regarding adversarial training. We chose to not recompute adversarial examples at every step largely because the L2 norms of the subtokens, which we use to generate adversarial examples, do not change quickly. Furthermore, since we find that the L2 norm is correlated strongly with frequency in the dataset, we expect that over time, subtokens that have a high L2 norm at one point will continue to have a high L2 norm. Thus, the generated adversarial examples would not change from step to step, and so, for simplicity, we compute them beforehand. However, we do agree that our evaluation of the adversarial training is preliminary and should be expanded. While our results suggest that adversarial training is effective with our technique, since the results are preliminary we will move them to the appendix.
>
> Thank you again for your helpful commentary and critiques; we are currently in the process of revising to address these concerns.

---

### Official Review · AnonReviewer2 · 2020-11-01
**An Adversarial attack strategy for code2seq model.**

**Rating:** 4
**Confidence:** 5

**Review:**

The paper proposed an Adversarial attack strategy for the code2seq model.
The authors observed that L2 distance between pre- and post-training embedding of a token varies significantly based on the token’s frequency. Based on that they devised strategies (black and white box) to perturb the sub-token. Here, they change the local sub-token names using three heuristics.

I like the simple approach to launch the adversarial attack, which shows a promising result. However, I have the following concern:

-	The paper lacks motivation. Are there any security implications of such attacks (e.g., malware classification, etc.)?
-	Only for one model (Code2Seq) the authors have tested their scheme.
-	Did not compare with other attacks that work on discrete domains (especially in NLP there are many attacks)
-	The three strategies proposed in the paper for perturbation. However, these strategies are quite ad-hoc (e.g., why suddenly choose 5 sub-tokens).
- The paper said a black-box attack (even in the title), but they reply on white box information showing that L2 distance of high-frequent tokens are more.

---

> ### Author Response · Authors · 2020-11-14
> **Thank you for your feedback**
>
> My co-authors and I would like to thank you for compiling this thoughtful review of our work, STRATA. We would like to share the following response to your comments:
>
> * You note that the section on motivation for the attack is under-developed; this critique is well-taken and in our revision, we include standard and malware-detection-specific motivations: The threat of adversarial examples poses severe risks for ML-based malware defenses, and introduces the ability of malicious actors to trick ML-based code-suggestion tools to suggest bugs to an unknowing developer. Thus, developing state-of-the-art attacks and constructing machine learning models that are robust to these attacks is important for computer security applications.
> * The code2seq model is state-of-the-art for models of code, and due to the fact that STRATA depends on training set token distributions and frequency differentials, we have no reason to believe that STRATA attack will be any less effective on other deep learning models of code or natural language. This fact justifies our choice of model. Nonetheless, in our revision, we expect to demonstrate the efficacy of our attack on at least one other model. We are currently working on attacking the Allamanis et al. convolutional attention network (Allamanis, 2016; http://proceedings.mlr.press/v48/allamanis16.pdf)
> * In response to your concern that our attacks are ad-hoc due to the choice of five as the number of concatenated subtokens, we want to clarify that we chose five because the code2seq model, with the configuration specified by the original authors, will ignore any subtokens after the fifth, and thus five is maximal and represents a “best-case” against this architecture. We will be sure to make this more clear in the revision to the paper.
> * We do discuss our results as they compare to the relevant discrete-space literature on adversarial models of code (Tables 6 and 7). However, we will bolster this section of the paper in a revision to make the comparisons more clear. We conducted a search through the NLP literature to find similar attacks, but we did not find any; perhaps unsurprisingly, as this mode of attack works best in the code setting where semantics are unaffected by variable replacement, whereas in the NLP setting the semantics can be greatly affected by a single word replacement if not chosen carefully (i.e. from a set of synonyms).
> * We show the results of a black-box attack using only the frequency of tokens in the training set (or, to be truly black-box, in a non-overlapping code dataset, as shown in the cross-dataset comparisons). The point is well-taken that we do not emphasize the black-box attacks as much as the title would suggest, and we will edit the title in response.
>
> In addition to answering the above critiques, we also plan to add more programming languages (such as Python) to our system so we can demonstrate more versatility; we have every reason to think that our attack strategy will be just as effective under those circumstances.

---

### Author Response · Authors · 2020-11-14
**Summary of response to reviewers**

Thanks to all our reviewers and their thoughtful commentary on our work, STRATA. Here we’ll summarize a few high-level points that we hope will be useful in a continuing discussion about the merits of this work:

* We believe that STRATA has a broad application to the field of machine learning in discrete spaces with large vocabularies. We are presenting the insight that subtoken frequency correlates with L2 norm and thus can be used to determine high impact subtokens; we do not believe that this technique is exclusive to code2seq or even models of code in general since this should be a property of the dataset distribution and training technique and not the model architecture. Skip-gram word embeddings have the same distribution (see Figure 1 of https://arxiv.org/pdf/1805.09209.pdf), and thus we believe that our technique should be applicable in the NLP setting (with modification; for example, replacement dictionaries might only be filled with high-impact synonyms). We believe that our results are fundamental to the dataset distribution and training process, and that they should generalize to other applications, so we want to make the community aware of this phenomenon. However, we do grant we could emphasize this point better in the writing.
* Since STRATA is gradient-free and does not (necessarily) require re-computation every step in the adversarial training process, our method is extremely inexpensive, whereas traditional gradient-based methods for developing adversarial examples are highly expensive, and are less effective. Therefore, it can represent an extremely easy hardening technique for all types of models operating in the discrete domain with a large vocabulary space. Further, we have shown that traditional adversarial training isn’t effective against our attack, which we believe to be a high impact contribution on its own.
* Nevertheless, we agree that more data and experiments can only help bolster our above points. We plan to test our attack against a different model, specifically the Allamanis et al. convolutional attention network (Allamanis, 2016).
* We also plan to test our attack on a code2seq model trained with the Python programming language.

Again, we appreciate your time and commentary. We are working on a revised version of the manuscript that emphasizes the importance of this technique better, and which is less tightly scoped.

We’ll add results and comments here as work on the revision progresses.

---

### Author Response · Authors · 2020-11-24
**Summary of revision**

We would like to thank all of the reviewers again for their helpful feedback. We have posted an updated version of the paper, and here we summarize the changes:
* We have changed the title to emphasize the fact that STRATA is gradient-free, and to remove from the title our claim that our attack is black-box attack, since the attack relies on having a similar dataset to the training dataset, and thus warrants the note that we provide in the body of the text about this point when we introduce the black-box version of the attack.
* We have added the results of applying our attack to a code2seq model trained on the Python150k dataset, as to confirm that our attack is not programming-language specific. Please see the updated Table 2 and Figure 2 in the paper. The baseline Python model is has an F1 score of 0.313 on the Python150k testing dataset and our attack lowers this F1 score to 0.153 for the embedding-L2-norm-based version of attack and 0.172 for the subtoken-frequency-based version of the attack. We have added accompanying discussion of this result in the paper in Section 4.1.
* Reviewers have pointed out that our adversarial training requires more evaluation and is thus preliminary, so we have moved the sections discussing adversarial training to the appendix, and added this to our future directions section in the conclusion.
* We changed the example of an adversarial attack on code to illustrate a better example of how a human would not be fooled by a change, yet code2seq fails. The example presents a function that computes the sum of a list of integers. Initially, code2seq predicts “sum” but when renaming the local variable “total” to “answer”, code2seq predicts “getInt”.
* We improve the discussion of our motivation for choosing to attack code2seq in the beginning of section 2.
* We rework the conclusion for clarity and to emphasize the limitations of STRATA and important future directions, and to tie our results to our hypothesis of how STRATA may be broadly applied to discrete domains with large vocabularies. We emphasize that we do not believe that STRATA is unique to code, but that this should be left for future work. We want to point out that while our study examines only code2seq, the phenomenon is based on observations of the training process and dataset distribution, and we want to bring to the attention of the broader community of our observations
* We clarify the Ramakrishnan et al. comparison and provide results of our attack on the entire testing dataset of java-small, rather than the subset of examples with local variables to perturb. We do not perturb examples without local variables. Nonetheless, we would like to note that while we evaluate our attack by perturbing local variable names, STRATA is a method concerned with identifying likely candidates for high impact tokens, and thus can be applied anywhere where tokens can be changed.

We believe that the changes we have made have improved the paper by addressing many of the reviewer’s concerns and have broadened the scope of the paper.

---

### Decision · Program_Chairs · 2021-01-07
**Final Decision**

**Decision:**

Reject

**Comment:**

The paper gives a gradient-free method for generating adversarial examples for the code2seq model of source code.

While the reviewers found the high-level objectives interesting, the experimental evaluation leaves quite a bit to be desired. (Please see the reviews for more details.) As a result, the paper cannot be accepted in the current form. We urge the authors to improve the paper along the lines that the reviews suggest and resubmit to a different venue.